

# Influence of Atmospheric Stratification on the Integral Scale and Fractal Dimension of Turbulent Flows.

**M. Tijera[1] , G. Maqueda[2], C. Yagüe[3]**

[1] Applied Mathematics Dpt. (Biomathematics). Complutense University of Madrid, Madrid, Spain.

[2] Astronomy, Astrophysics and Atmospheric Science Dpt. Complutense University of Madrid, Madrid, Spain.

[3] Geophysics and Meteorology Dpt. Complutense University of Madrid, Madrid, Spain.

Correspondence to: M.Tijera (mtijera@fis.ucm.es )

**Abstract**

In this work the relation between integral scale and fractal dimension and the type of stratification in fully developed turbulence is analyzed. Integral scale corresponds to that in which energy from larger scales is incoming into turbulent regime. One of the aims of this study is the understanding of the relation between the integral scale and the Bulk Richardson number, which is one the most widely used indicators of stability close to the ground in atmospheric studies. This parameter will allow us to verify the influence of the degree of stratification over the integral scale of the turbulent flows in the Atmospheric Boundary Layer (ABL).The influence of the diurnal and night cycle in the relationship between the fractal dimension and integral scale is also analyzed. Fractal dimension of wind components is a turbulent flow characteristic as it has been shown in previous works, where its relation to stability was highlighted. Fractal dimension and integral scale of the horizontal (u`) and vertical (w`) velocity fluctuations have been calculated using the mean wind direction as framework. The scales are obtained using sonic anemometer data from three elevations 5.8 m, 13 m and 32 m above the ground measured during the SABLES- 98 field campaign.  In order to estimate the integral scales a method that combines the normalized autocorrelation function and the best gaussian fit ($R^2 \geq 0.70$) has been developed. Finally, by comparing,





at the same height, the scales of u` and w` velocity components it is found that almost
always the turbulent flows are anisotropic.
**1    Introduction**
The aim of this paper is to bridge the considerable gap that exists between the fractal
dimension and the integral scale. The size of the integral scale of the horizontal and
vertical components and fractal dimension of wind velocity near the earth's surface in
boundary layer are determined. Also, these magnitudes are compared between them and
versus other parameters such as the Bulk Richardson number. It is assumed that the
turbulence is the primary agent that causes changes in the boundary layer. In turbulent
flows it is observed that time series of meteorological variables  as wind velocity,
temperature, pressure and other atmospheric mechanical magnitudes fluctuate in a
disordered way with peaks extremely sharp and irregular space and time variations.  The
complicated  nature of these series indicates that the motion of the air is turbulent. If we
take a good look at the variety of fluctuations of different periods and amplitudes
observed in them we could explain the complicated structure of turbulence. The
irregularity of the time series obey to the existence of different size and time scales and
also to the nonlinear transfer of energy that exists between them in the turbulent flows
(Monin and Yaglom, 1971).
The irregular behavior of these flows is also due to waves and turbulence that are often
superimposed on a mean wind (Stull, 1998). If we filter the mean wind and waves in the
appropriate range we will only have turbulence. Some previous works present results
about this procedure (Tijera et al., 2008). In this paper we have carried out the necessary
transformation to get the mean wind series in short intervals, namely 5 minutes. We
filter horizontal and vertical mean wind velocity obtaining the time series of
fluctuations of the velocity in both directions ($u' = u - \bar{u}$ , $w' = w - \bar{w}$).
When we observe these time series such as wind velocity, they vary in an irregular
shape and in spite of their complexity presents a self-similarity structure (Frisch,1995) .
This is a common property of the fractals, so that wind velocity could be considered as a
fractal magnitude. The modern physical notion of fractals is largely known due to
Mandelbrot (1977, 1985), but the mathematical notion of curves lines or sets having
noninteger dimensions is much older (Hausdorf, 1919, Besicovitch, 1929). An analysis



that compare the Haussdorff dimension and Kolmogorov capacities of self-similar
structure with non integer fractal dimensions (Kolmogorov capacity or box counting
dimension) was presented by Vassilicos (Vassilicos and Hunt, 1991).  The wind
velocity versus time are irregular curves of this type, with noninteger dimensions. These
values correspond to the fractal dimension. A way of measuring the complexity of these
series is by means of fractal dimension. The Fractal Dimension of wind components is a
characteristic of turbulent flow as it has been shown in previous works where its
relation to stability was highlighted (Tijera et al.,  2012)
In this paper the integral scale of u` and w` component are compared. The scales are
calculated using sonic anemometer data from three elevations 5.8 (~ 6), 13 and 32 m
above the ground at the main tower site of the Sables 98 field campaign. Turbulent
motion of the atmospheric fluxes that occurs from the smallest scales are usually
defined as the scale at which the motion dissipates into heat due to the viscosity of the
fluid until the larger scales that correspond the integral scale. The integral scale can be
defined in several ways: the larger scale of the flow, the scale above which the Fourier
transform has a slope inferior a -5/3 slope, as which the turbulent kinetic energy (TKE)
is maximum. Micrometeorological studies have found integral scale varying in a huge
range, from around a hundred to a thousand meters (Teunissen, 1980, Kaimal and
Finningan, 1994).
We study the anisotropy of the turbulent atmospheric flows in these scales comparing
integral scale of fluctuations of the velocity component along of the mean wind
direction and the vertical component at three different levels above the ground (5.8 m,
13.5 m, 32 m).

## 25    2    Theoretical background

The irregular behavior of the atmospheric turbulent fluxes in the boundary layer at large
Reynolds number leads us to be interested in calculating their fractal dimension. Fractal
dimension could help us to classify the irregularity of these flows. The more irregular
the flow the greater its fractal dimension. Turbulent flows are characterized by the
formation of many eddies of different length scales. Theses irregularities are due to the
superimposition of eddies of different sizes and it is related with a broad range of scales
which exist in turbulence. These scales vary from the smallest scales as dissipative scale
to larger scale as integral scale. This paper is concerned with the analysis of the



relationship between the integral scale and fractal dimension. As well as the relationship
between the integral scale with the Bulk Richardson number, a turbulent and stability
parameter which is used to characterize the degree of stratification in the atmosphere.
In this section we describe the methodology applied to calculate the fractal dimension
and the integral scale. The estimation of the fractal dimension of time series has been
the most commonly used criteria to measure their chaotic structure, there exist different
works in that direction (Grassberger and Procaccia, 1982, Shirer et al, 1997). One of the
methods most commonly used to estimate to fractal dimension of atmospheric flows has
been the mean slope method through box-counting dimension using mean slopes of the
graph of ln N (L) versus ln (L) for small ranges of L, where N(L) is the number of the
boxes of side L necessary  to cover the different points  that have been registered in the
physical space (velocity-time) (Falconer, 2000, Peitgen et al., 2004). As $L \rightarrow 0$ then
N(L) increases, N meets the following relation:
$$N(L) \cong k L^{-d} \qquad\qquad (1)$$
The value  d is the box-counting dimension that is an approximation of the Hausdorff
dimension  and is calculated approximately by  means of  least-square-fitting of  the
representation  of log N(L) versus log L obtaining the straight line regression given by
the following equation:
$$\log N(L) = \log k - d \log L \qquad\qquad (2)$$
The fractal dimension d will be given by the slope of this equation as is shown in the
Fig.1.
In this paper we focus on calculating the integral scales for the mean wind direction $u'$
as horizontal component and vertical velocity as component $w'$ and we studied their
variations with respect to the fractal dimension and with the Bulk Richardson number, a
turbulent parameter of stability.
These integral scales have been estimated using the normalized autocorrelation function
and a Gaussian fit.  The velocity autocorrelation function as a function of $\tau$  (lags
number) for  $u'$ component is:
$$R(\tau) = \frac{\overline{u'(t)u'(t+\tau)}}{\overline{u'^2(t)}} \qquad\qquad (3)$$
Integral time scale is:





$$T_L = \int_0^\infty R(\tau)d\tau \approx \int_0^{\tau'} R(\tau)d\tau \qquad (4)$$
The integral time scale provides a measure of the scales of eddies in the x direction of
a flow field. In the Eq. (2) we observed that $\tau'$ denotes the last lag in the data series. In
boundary layer observations this time scale can be related to a length by multiplying the
mean wind velocity by time scale. This requires the assumption of frozen turbulence
known as Taylor's hypothesis (Panofsky and Dutton, 1984). The integral length scale
can be defined as:
$$\lambda = \bar{v}T_L \qquad (5)$$
The used method is based on Gaussian fit of the normalized autocorrelation function
R($\tau$) and we calculated the value of $\tau$ that verifies the following equation:
$$\tau - \int_0^\tau R(\tau)d\tau = \int_\tau^{\tau'} R(\tau)d\tau \qquad (6)$$
The Fig. 2 shows the Gaussian fit for an example of a data series of wind velocities with
$\tau$ that verifies Eq. 6. This value allows us to calculate the integral time scale
multiplying it by the time interval between each lag.

## 3  Description of  Data

The data set was recorded in the Research Centre for the lower Atmosphere (CIBA in
the Spanish acronymus), located in Valladolid province (Spain) and were measured in
the experimental campaign Sables-98. This research centre was set up primarily to study
the atmospheric boundary layer. The campaign took place from $10^{th}$ to $27^{th}$ September
1998 (Cuxart et al., 2000). This experimental site is a quite flat and homogeneous
which forms a high plain of nearly 200 $Km^2$, surrounded by crop fields and some small
bushes strewn over ground. Duero river flows along the SE border of the high plain.
The synoptic conditions during the period of study of eight consecutive days (from 14
to 21 September) were controlled by a high pressure terrain system which produces
thermal convection during the diurnal hours and from moderate to strong stable
stratification during the nights.
Here we analyze sonic anemometer data installed at 5.6 (~ 6), 13 and 32 m, these data
set are five minute series. These series have been obtained once we have carried the



necessary transformation to get the mean wind velocity series in short periods of 5
minutes. At a rate of 20 data points per second, sonic anemometers can resolve integral
scales between about 10 m to 2000 m of u' horizontal component and 1 m to 1000 m of
the w' vertical component, depending on the height in which the sonic anemometer is
positioned and at the wind speeds typically measured in the Sables-98 experiment.
**4  Results**
**4.1 Fractal Dimension, integral scale and stability of stratification.**
In this paper we analyze the influence of stability of stratification on fractal dimension
and integral scale. Different   atmospheric surface-layers data are separated into thermal
and dynamics stability classes based on a dimensionless parameter such as the Bulk
Richardson number $Ri_B$. This parameter represents the ratio of the production or
destruction of turbulence by buoyancy and by wind shear strain that is caused by
mechanical forces in the atmosphere:

$$Ri_B = \frac{g}{\overline{\theta}} \frac{\Delta\overline{\theta}\Delta z}{(\Delta\overline{u})^2} \qquad (7)$$

where $g$ is the gravity acceleration and $\overline{\theta}$ the average potential temperature at the
reference level, the term $\dfrac{g}{\overline{\theta}}$ is known as the buoyancy  parameter. $Ri_B$ is positive for
stable stratification, negative for unstable stratification and approximate zero for neutral
stratification (Arya, 2001). The way to calculate this number is described next:
1.    Calculation of the mean potential temperatures at height $z = 32$ m, and close to the

21        surface $z = 5.8$ m, namely  $\overline{\theta_{32}}$  and   $\overline{\theta_{5.8}}$  respectively. Being  $\Delta\overline{\theta} = \overline{\theta_{32}} - \overline{\theta_{5.8}}$ .

2.    Obtaining of  $\overline{u_z}$  the mean wind velocity module at the height $z = 32$ m and $z = 5.8$

23        m, denoted by  $\overline{u_{32}}$ and  $\overline{u_{5.8}}$  respectively, where $\Delta\overline{u} = \overline{u_{32}} - \overline{u_{5.8}}$ .

Once the values of  $\Delta\overline{\theta}$ ,  $\Delta\overline{u}$  and $\Delta z$ have been obtained by means of Eq. (7) we
calculate the Bulk Richardson number in the layer between 32m and 5.8m.
In Fig. 3 we present the variation of the fractal dimension of the u' horizontal
component of the velocity fluctuations along time at the three considered heights: 5.8 m,





13 m and 32 m. The behaviour of these variations is similar at the three heights. The w'
component fluctuation presents an analogous behaviour. The fractal dimension values
are in a range between 1.30 to nearly 1.00. We have found that during the diurnal hours
the fractal dimension is bigger than at night (Tijera, 2012). We have no theoretical
reason to explain this result, but a possible explanation of why this happens could be
that fractal dimension is related with atmospheric stability and with the intensity of
turbulence. It is well known that the intensity of turbulence grows up as solar radiation
increases, producing instability close to the ground, mainly in the hours of noon.
Therefore, one of the possible reasons of the increase of FD is the instability of the
turbulent flow.  In the other hand, during the nights a strong atmospheric stability
usually exists, so the fractal dimension is usually smaller than during the diurnal hours.
Thus, stable stratification decreases the fractal dimension.
In Fig. 4 it is observed a different behaviour in the integral scale for u` horizontal and
w` vertical component, it is not clear how it influences on the diurnal cycle. Sometimes
during the night and the noon the scales increase or decrease in a stochastic way and this
is due to the chaotic structured motion of the atmospheric fluid that we refer as
turbulence. It is observed in the own integral scale many eddies of many different
lengths. The integral scale for u` component varies between around 100 m on their
minor scales, until above 1500 m for its mayor scales. The integral scales for    w`
component are slightly lower than u` component as it is indicated in Fig. 4. It is shown
that these vertical scales can reach sizes between a few tens of meters until 1000  m in
some occasions. It is observed in each component that the greater the height the greater
the integral scale. Usually, scales at 32 m are greater on average than those at 13 m and
the latter higher than at 5.8 m
Although the mental model of turbulence as eddies of various sizes is useful, it is
difficult to obtain a correlation between the integral scale and fractal dimension in the
atmosphere if we consider values throughout the whole day. However, it is much easier
to find a relationship between the integral scale and fractal dimension of horizontal and
vertical components of the wind velocity if we separate the hours of the day and night,
and hence analyze the influence of diurnal and night cycle over these parameters.
Daylight hours are from 6-18 UTC and the night from 18-6 UTC. These data set are
analyzed in the three studied heights.  Fig. 5 shows the variations of the integral scale
versus fractal dimension at the level of 5.8 m for horizontal component. As it can be



appreciated in  Fig.  5 in the diurnal hours the average values of the integral scale versus
the fractal dimension can be adjusted to the straight regression line given by the linear
equation that appears on the top left of the graph. During those hours these values of the
integral scale increase from a few tens of meters until 400 m with increasing values of
the fractal dimension until 1.25. During the nights the average values of the integral
scale decreases with the increase in the fractal dimension.  These values also fit a
straight regression line as it is indicated in Fig. 5. One of the possible explanations for
this behaviour is that during the diurnal hours the average values of the integral scale
increase due to the unstable stratification. During the nights, the existence of the stable
stratification decreases the integral scale with an increase in fractal dimension until the
approximate value of 1.2.  This tendency appears also in the other two heights, at 13 m
and at 32 m as it is shown in Fig. 6. Although during the diurnal cycle at 32 m the linear
fit it is not so evident, the maximum scales are in the 1.15-1.20 range of the fractal
dimension as it is illustrated in  Fig. 6 (c).
For the vertical component in the three studied levels, the behaviour is slightly different.
The averages values have fitted a quadratic function as it is indicated in  Fig. 7. During
the diurnal hours the averages values of integral scale reach maximum scales  around
the value of the fractal dimension of 1.15 at the three heights. From this value the
integral scale decreases when fractal dimension increases.These maximum integral
scales depend on the height. At the level of 5.8 m their sizes reach 50 m in average and
the scattering of the values shows higher values that could reach 100 m. At the level of
13 m the values are about 100 m, the dispersion of these scales can reach sizes of 200 m
and at the height of 32 m their larger average scales are approximately around 200 m
and due to the variances of the data set could reach sizes of 400 m. From the value of
the fractal dimension value of 1.15, the scales decrease until a few meters.
Throughout the night the average values of the integral scales decrease with the increase
of the fractal dimension in a parabolic way as it is indicated in Fig. 7. This happens due
to the stable stratification that occurs at nights. This behavior during the diurnal and
night hours for w` component of the integral scale is similar to the results obtained for
u` component, although the fits of averages values are parabolic and not linear. In all
these cases our $R^2$ values and confidence levels  are high as it is indicated in Fig. 7.




## 4.2 Relationship between integral scale and Bulk Richardson number

Among the numerous parameters existing to characterize the degree of stratification in the atmosphere we will use the Bulk Richardson number. The interpretation of this number has already been mentioned in the previous section. Here, we analyze how the integral scale of each one of the u` horizontal and w` vertical components varies with the Bulk Richardson number in diurnal and night cycle in the studied period. These results are shown in Fig. 8 for horizontal component and in Fig. 9 for vertical component. During the daylight hours appear the three kinds of stratification: unstable, neutral and stable as it is shown in the three graphs on the left side of Fig. 8 and the Fig. 9, each one corresponds to the different heights. In the stratification unstable and neutral the integral scales are higher than the integral scales under the influence of the stable stratification. At 5.8 m for the horizontal component these scales vary between 200 m and values slightly higher than 400 m and in the case of neutral stratification could increase until 600 m. This same behaviour occurs in the other two studied heights 13 m and 32 m although their scales are slightly higher as it is illustrated in Fig. 8. During the nights it is observed the biggest stability due to positive values of the Bulk Richardson number.

 The same results are obtained for the integral scales of vertical component, although their sizes are smaller. At 5.8 m during the diurnal hours the average values reach about 50 m and during the night hours their values are below 50 m. At 13 m and 32 m in the diurnal hours the average values could reach about 150 m and 200 m and at the night hours are below 100 m and 200 m respectively.

## 4.3 Analysis of the anisotropy with the integral scale

In the last section we study the relationship between the integral scales of the horizontal and vertical components at different heights: 5.8 m, 13 m and 32 m. In  Fig. 10 we represent the integral scale of u` component versus the integral scale of w` component at three studied heights and we find linear relations with the averages values of these scales. All integral scales measured during the period of study from 14 to 21 of September appear in this figure.  The linear fits obtained are acceptable, with high $R^2$ values at 13 m and 32 m, as it is indicated  in  Fig. 10. The linear regression appears on the top left of each graph: at 5.8 m $L_{intu}(5.8\ m) = 1.46\ L_{intw}(5.8\ m) + 178$,  at   13 m



$L_{intu}(13\ m)=0.957\ L_{intw}(13\ m)+275.6$ and at 32 m $L_{intu}(32\ m)=0.646 L_{intw}(32\ m)\ +370$,
being $L_{intu}$ and $L_{intw}$ the average values of the integral scale for horizontal and vertical
component respectively.
The data in Fig. 10 appear quite scattered and the average values could be representative
to find relationships between these scales. This scatter is due to the large number of
uncontrolled variables, nonlocal disturbance, the presence of waves, horizontal
inhomogeneity, low frequency disturbances, etc. These graphs are showing that the
scale measured at 32 m is nearly always larger than the integral scale measured at 5.8 m.
On the basis of the results obtained, we find slight differences between these
components, thus there is anisotropy in atmospheric turbulent flows. In isotropic
turbulence the integral scales of both components should be the same at the same
height. Only under certain conditions and over limited scales is isotropy a property of
turbulence in the stratified atmosphere (Thorpe, 2005).

## 5   Conclusions

In this paper algorithms have been developed to calculate the fractal dimension and the
integral scale using wind velocity data from the convective boundary layer. We present
some results related to the time evolution of both fractal dimension and integral scale.
As well as how the day and night cycle affects the relationship between the fractal
dimension and integral scale and their behavior versus Bulk Richardson number. The
different levels of stratification help us to understand the relations between the fractal
dimension and integral scale. The stratification in the atmosphere has showed some
degree of the influence with the most of the integral scales. The main conclusions of
this study are as follows.
Although all data appear quite scattered in this work, the averages values of these
magnitudes show interesting results. During the diurnal hours the averages values of the
integral scale of the horizontal component increases with the increase in fractal
dimension until around 1.25 at 5.8 m and 13 m height. At these heights we have found
linear fits between these magnitudes with high coefficients of correlation. While at 32 m
the linear fit is not so evident, the maximum scales are in the 1.15-1.20 range of the
fractal dimension. One of the possible explanations for this behaviour is that during the
diurnal hours the average values of the integral scale increase due to the unstable





stratification. During the night hours the average values of the integral scale decreases
with the increase in the fractal dimension.  These values also fit a straight regression
line at the three analysed heights. During the nights the existence of the stable
stratification decreases the integral scale with an increase in fractal dimension until the
approximate value of 1.2.
For the vertical component of the integral scale the results are similar, even though with
slight differences. The averages values have fitted a quadratic function. During the
diurnal hours the averages values of integral scale reach maximum around the value of
the fractal dimension  of 1.15 in the three heights. From this value the integral scale
decrease with the increase of the fractal dimension until a few meters. The different
degree of stratification along diurnal hours will be reflected in that different behaviour
from the value of 1.15. At nights when stability is normally major the integral scale
decrease with increasing the fractal dimension of a parabolic way
In the unstable and neutral stratification the integral scales are higher than the integral
scales under the influence of the stable stratification.
To characterize the anisotropy of turbulent flows we have used the comparison of
integral scales of horizontal and vertical component showing that the scale of u`
component is almost always larger than the scale of the w` component at the same
height.
**Acknowledgements**
This research has been funded by the Spanish Ministry of Science and Innovation
(projects CGL2009-12797-C03-03). The GR35/10 program (supported by Banco
Santander and UCM) has also partially financed this work through the Research Group
"Micrometeorology and Climate Variability" (No 910437). Thank to participant teams
in SABLES-98 for the facilities with the data.

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





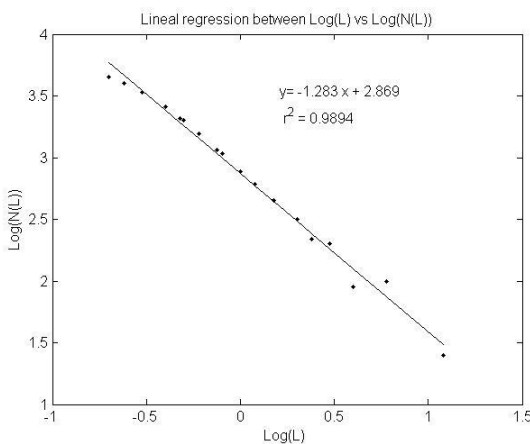

**Figure 1**.Example of linear regression between number of not empty boxes  and length
side of the box. The slope (*d*) is the fractal dimension of the w` component, *d* =1.28 $\pm$
0.03 for a example of the w$^{'}$ component of the wind velocity.
I

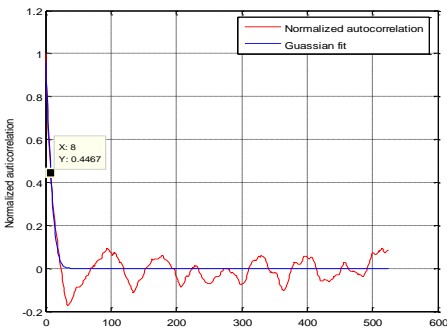

**Figure  2.** Gaussian fit for a data series of wind velocities  *u* ' component that allows us
to calculate the integral scale.





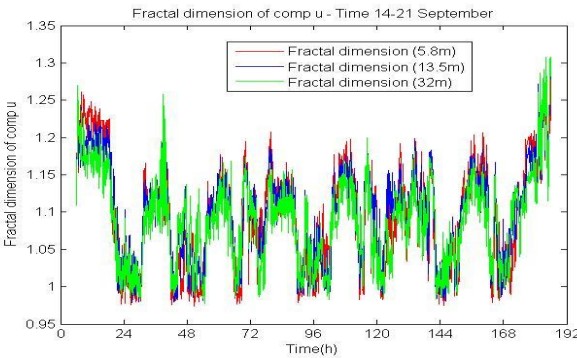

**Figure 3**. Variation of the fractal dimension versus time for the u' component
fluctuation at the three heights, showing the influence of the diurnal cycle.

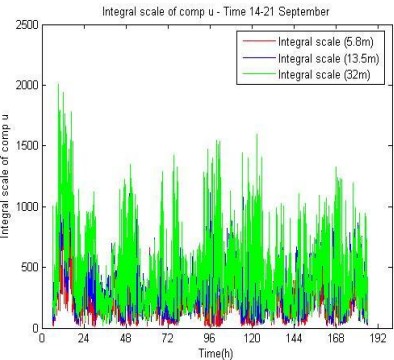
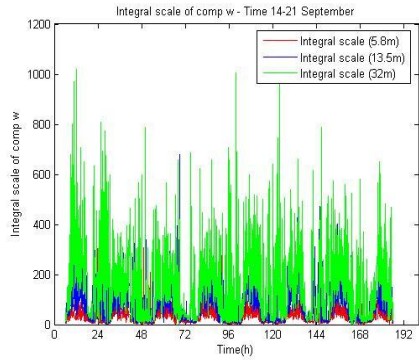

**Figure 4.** Variation of the integral length scale of horizontal and vertical components
versus time at the three heights.



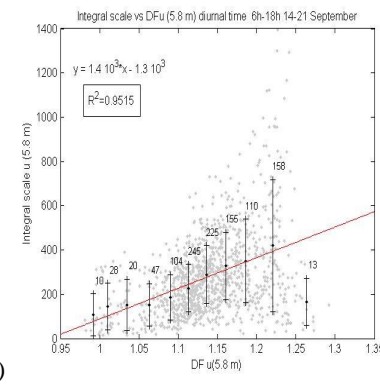
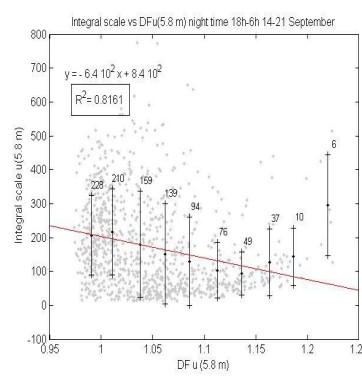

(a)                                                (b)

**Figure 5.** Variations of the integral scale versus the fractal dimension of u` component
of the wind velocity at 5.8 m. (a) diurnal hours 6h-18 h, (b) night hours 18 h - 6h. On
the top left of the each graph it is indicated  the linear regression of the averages values
(a) $L_{intu}(5.8\ m)= 1.4\ 10^3\ DF_u(5.8\ m) - 1.3\ 10^3$  (b) $L_{intu}(5.8\ m)= -6.4\ 10^2\ DF_u(5.8\ m) +$
$8.4\ 10^2$, being $L_{intu}$ and $DF_u$  the integral scale and fractal dimension for u component
respectively.



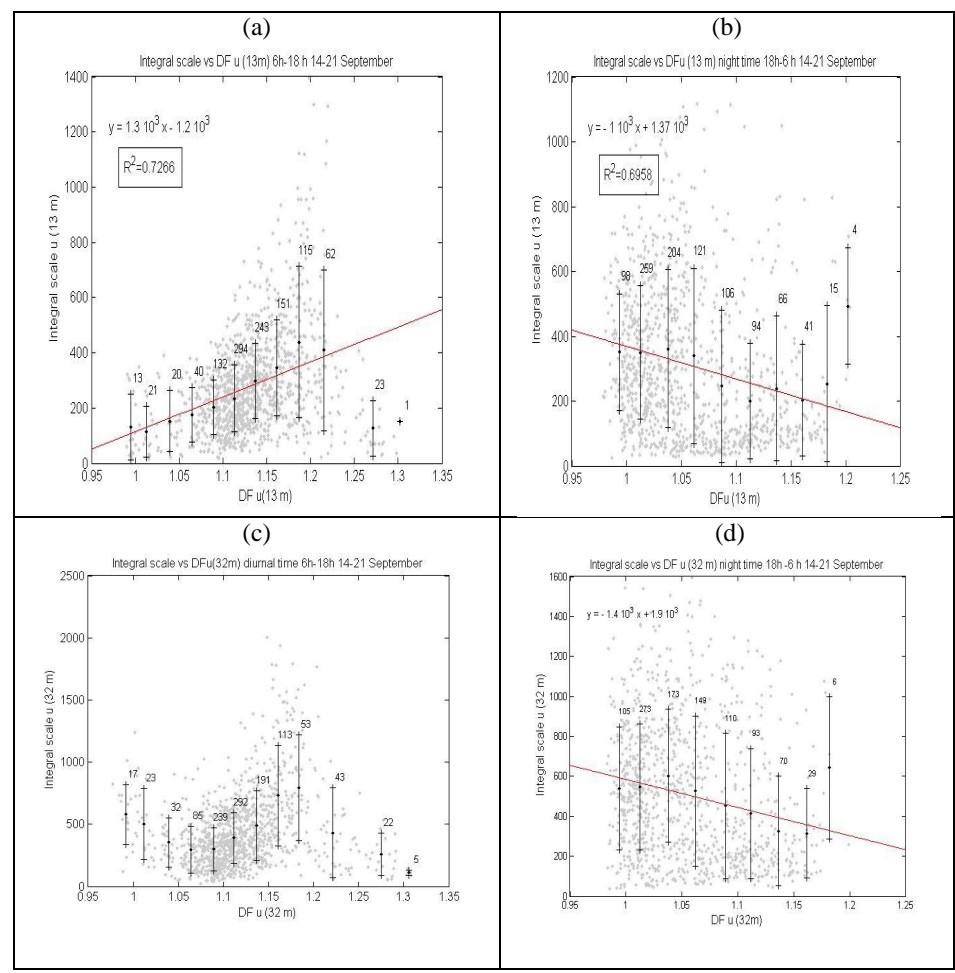

**Figure 6.**   Variations of the integral scale versus the fractal dimension of the u`
component at 13 m and 32 m. (a) and (c) diurnal hours, (b) and (d) night hours. In the
same manner that in the figure 5 the linear fits are: (a) $L_{intu}$(13 m)= 1.3 $10^3$ $DF_u$(13 m) -
1.2 $10^3$ , (b) $L_{intu}$(13 m)= -1 $10^3$ $DF_u$(13 m) + 1.37 $10^3$, (d) $L_{intu}$(32 m)= -1.4 $10^3$ $DF_u$(32
m) + 1.9 $10^3$







**Figure 7.** Variations of the integral scale versus the fractal dimension of the w`
component at 5.8 m, 13 m and 32 m. (a), (c) y (e) diurnal hours, (b), (d) y (f) night.
hours. The fits to a quadratic function of the averages values appear on the top left of
the each graph, being y variable $L_{intw}$ and x variable $DF_w$.





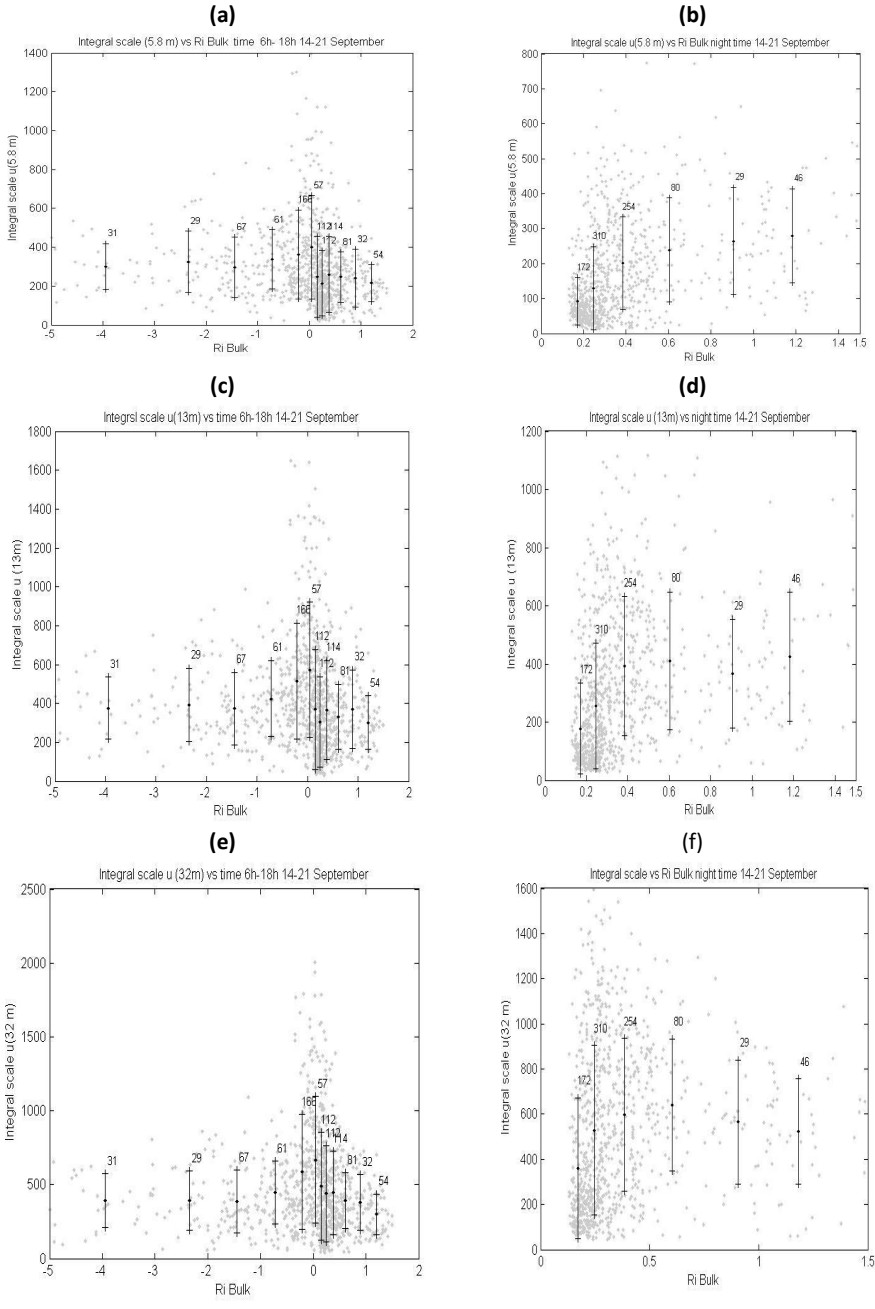

**Figure 8.** Integral length scales of u` component plotted against the Bulk Richardson number
at 5.8 m, 13 m and 32 m. (a), (c) y (e) diurnal hours, (b), (d) y (f) night hours.





**(a)**

**(b)**

**(c)**

**(d)**

**(e)**

**(f)**

1    **Figure 9.** Integral length scales of w` component plotted against the Bulk Richardson
2    number at 5.8 m, 13 m and 32 m. (a), (c) y (e) diurnal hours, (b), (d) y (f) night hours





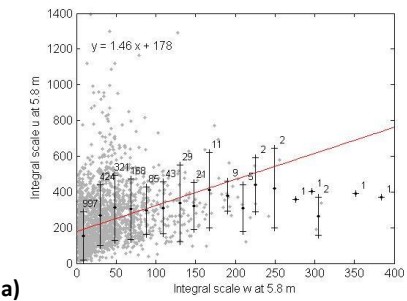

(a)

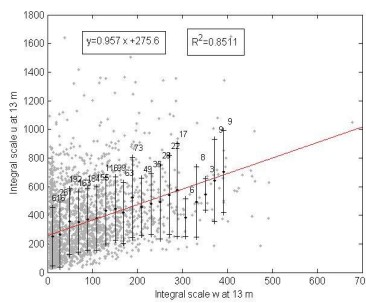

(b)

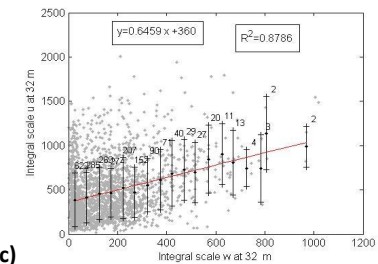

(c)

**Figure 10.** Comparison of integral scales of u` component and w` component (a) at 5.8 m, (c) at 13 m (c) at 32 m, showing that the averages values of theses scales fit to the linear regression indicated on the top left each graph. Data set appear as a cluster around the straight line.

