# Peer review of "Influence of Atmospheric Stratification on the Integral"

_Nonlinear Processes in Geophysics, 2015_

## Referee Comment (RC1) · Anonymous Referee #1 · 11 Apr 2016

Report for Nonlin. Processes Geophys.

Manuscript number: npg-2015-77

Title: "Influence of Atmospheric Stratification on the Integral Scale and Fractal Dimension of Tubulent Flows"

Authors: "M. Tijera, G. Maqueda, C. Yague"

This paper investigates the possibility of correlations between the integral scale of stratified turbulent flows in the boundary layer and parameters characterizing topological features of the wind velocity field, such as the fractal dimension, and its stability properties studied through the Richardson number. Using data of the SABLES-98 campaign, at moderate temporal resolution, authors present here a series of results which

provide interesting insights on the dynamics of the atmospheric flow in the site under study. The outcome of the paper is relevant for the audience of Nonlin. Processes in Geophysics and I am leaning to suggest it for the publication after a revision of the text along the lines detailed in the following. Such a revision is necessary to achieve a clear understanding of the validity of the results presented and to make them available to a large swat of readers.

**1 line 5-6 page 2. The text says: "The aim of this paper is to bridge the considerable gap that exists between the fractal dimension and the integral scale"**

This sentence does not make much sense in this form and it should be rephrased. Perhaps the authors mean to say that there is a lack of investigations on the connection between the integral scale and the fractal dimension as estimated in turbulent flows. Please clarify it modifying the text accordingly.

**2 line 22-23 page 2. The text is not clear, is it a 5 minutes running average used to remove the mean field? This point needs to be better explicated in the text.**

**3 line 11-14 page 3. The sentence within this lines is not at all clear, please rephrase it.**

**4 line 2 page 4. The Richardson number provides a measure of how a turbulent flow is prone to develop instabilities. I thus disagree with referring to this parameter as a number that characterize the degree of stratification in the atmosphere. This definition (used in many places in the paper) is misleading and it is very important to modify it in the text not to drive the wrong message that stratified flows cannot be unstable. Indeed the Richardson number is used to identify unstable regimes also in strongly stratified flows and strong instabilities (and turbulence) can develop in the direction of the gravity, characterized a vertical Froude number of order $\sim 1$ as described in Billant & Chomaz, "Self-similarity of strongly stratified inviscid flows," Phys. Fluids 13, 1645–1651 (2001).**

**5 line 22 page 4. The text says: ".. the integral scale of the mean wind direction u'".**

Here the authors probably means the horizontal direction but the way this is written sounds wrong. Please rephrase it.

**7 line 1-2 page 6. See comment ## 2 and please specify better the way the data have been processed. This is important for the readers to be able to reproduce (and therefore validate) the results presented in the paper.**

**8 line 3-4 page 6. Authors claim they resolve vertical integral scales from 1m to 1000m, which puzzles me a bit. Indeed, I am wondering how is it possible to detect (using the Taylor hypothesis) vertical scales larger than the hight(s) at which the sonic anemometer are placed? An answer to this point should be included in the report and and a compelling explanation integrated in the text.**

**9 line 20 page 6. Details on how the potential temperature has been estimated should be included in the text.**

**10 line 12 page 7. The text says: "Thus, stable stratification decreases the fractal dimension."**

Authors acknowledge they cannot provide rigorous arguments to explain the variation of the fractal dimension with the height. As a consequence the statement above sounds a bit too strong, unless they propose some solid argument to support it.

**11 line 14-27 page 7. The text within this lines is poorly written and its meaning is a bit confusing. I suggest to re-written it from scratch and perhaps make it a bit more concise.**

**12 There is a long standing debate in literature on whether an inverse cascade occurs in stably stratified anisotropic flows (with or without rotation). The inverse cascade mechanism, if any, might also be responsible for the growth of the integral scale in the stratified atmosphere. I thus suggest the authors to address this point, perhaps when they re-arrange the text (line 14-27) as indicated in the previous comment (## 11). On this note/topic I suggest to cite these two papers:**

L. Smith and F. Waleffe, "Generation of slow large scales in forced rotating stratified turbulence," J. Fluid Mech. 451, 145–168 (2002).

R. Marino, P.D. Mininni, D. Rosenberg, and A. Pouquet, "Large-scale anisotropy in stably stratified rotating flows". Phys. Rev. E 90, 023018 (2014).

**13 Conclusions should definitely be rearranged from line 16-24 to achieve a better clarity of the text.**

**14 Units are missed in some plots, for instance in figure 2 "tau" is indicated but it is not clear whether the lag is given in seconds, minutes or hours. Please double-check all the figures and add the units (on the axes/legend) when it feels needed.**

———————————————

---

## Referee Comment (RC2) · Anonymous Referee #2 · 18 Apr 2016

The manuscript reports experimental measurements of velocity components recorded with a sonic anemometer in the Atmospheric Boundary Layer (ABL) over homogeneous terrain. The goal of the work is to study the behaviour of the wind fluctuations integral scale, and so-called fractal dimension, at changing the atmospheric stratification. Temporal resolution is moderate, but this is not a problem in principle since the authors do not focus on rapidly varying observables.

Let me first recall that over the last 10 years, the statistical characterisation of velocity turbulent fluctuations in the atmosphere is a widely studied topic. Numerous results from experiments and from numerical simulations are published in the literature.

The fractal dimension, or box-counting dimension $D_0$, is a statistical property of turbulent signals, providing a measure of the degree of roughness, and it is thought

to be a 'universal' property of it (see also many text books on the topic of the relation among the generalized fractal dimensions and the singularity spectrum of a turbulent signal).

This means, in particular, that it should not depend on the value of the integral scale of the flow, something that is clearly not universal but depends on the geometry of the system, and on its stability. If the authors differently think the such a relation exist, they should give a solid basis to their working hypothesis. A linear regression can be applied to any pair of observables.

The computation the fractal dimension is in principle an easy task, provided there exists an extended range of scales where there is scaling. The authors extract the fractal dimensions at fixed heights in the ABL with no discussion about the evolution of the mixed layer, and kind of flow they expect to measure at different height depending on the stratification. For example: similar to homogeneous and isotropic 3D turbulence during convection, similar to homogeneous stratified flow at night, etc.. Even if the diurnal cycle is never really stationary, there are stages of quasi-stationarity when statistical properties can be assessed.

Which fractal dimension do the authors expect to measure for daily convection and nightly stratified flows? There is no theoretical frame of reference in their analysis, just measurements.

Additionally to this major problems, the paper contains many "descriptive" sentences, whose necessity is at least doubtful, while the quantitative part on the analysis and discussion is very very weak.

Finally, I find that the manuscript contains limited novelty with respect to Ref. "Tijera M., Maqueda G.,Yaque C., and Cano J.: Analysis of fractal dimension of the wind 5 speed and its relations with turbulent and stability parameters , Intech, Fractal Analisis 6 and Chaos in Geosciences, 29-46, 2012"

For all these reasons, I think that the paper is not suitable for publication.

---

## Author Comment (AC1) · 4 Aug 2016

**Response to the Referee #1**

This paper investigates the possibility of correlations between the integral scale of stratified turbulent flows in the boundary layer and parameters characterizing topological features of the wind velocity field, such as the fractal dimension, and its stability properties studied through the Richardson number. Using data of the SABLES-98 campaign, at moderate temporal resolution, authors present here a series of results which provide interesting insights on the dynamics of the atmospheric flow in the site understudy. The outcome of the paper is relevant for the audience of Nonlin.Processes in along the lines detailed in the following.

We thank the referee for his positive reviews and constructive comments for our paper. We greatly appreciate the fact that our work is deemed acceptable for publication in NPGD with some revisions. In the following, we respond point by point to the referee 1 remarks ( in blue).

**1 line 5-6 page 2. The text says: "The aim of this paper is to bridge the considerablegap that exists between the fractal dimension and the integral scale"**

We agree with referee #1 toclarify this sentence, and have it replaced by the following in this paper:

The aim of this paper is to investigate the possible correlations between the integral scale of the turbulent stratified flows in the Boundary Layer and parameters charactering topological features of the wind velocity field, such as fractal dimension and its stability properties studied through the BulkRichardson number. We are aware that there is a lack of investigations between the integral scale and fractal dimension.

**2 line 22-23 page 2. The text is not clear, is it a 5 minutes running average used toremovethe mean field? This point needs to be better explicated in the text.**

We decided to change the sentence line 23-24 page 2 " In this paper we have carried out the necessary transformation to get the mean wind series in short intervals, namely 5 minutes ". It is not a 5 minutes running average.

In this work, the series of wind velocities in the three directions x, y and z recorded by the anemometer are divided in series of five minutes length. Each of these series applies the necessary transformation to get the mean wind series (u horizontal component) and vertical velocity (w vertical component)(Kaimal and Finningan, 1994).During 5 minutes the average values of the magnitudes in study remain constant.

**3 line 11-14 page 3. The sentence within these lines is not at all clear, please rephrase it.**

We rephrased the sentence line 11-14 page 3 by:

Turbulent motion of the atmospheric fluxes occurs through a broad range of scales from the smallest scales that are usually defined as the scales at which the motion dissipates into heat due to the viscosity of the fluid until the larger scales corresponding to the integral scale.

**4 line 2 page 4. The Richardson number provides a measure of how a turbulent flowis prone to develop instabilities. I thus disagree with referring to this parameter as a number that characterize the degree of stratification in the atmosphere. This definition (used in many places in the paper) is misleading and it is very important to modify it in the text not to drive the wrong message that stratified flows cannot be unstable. Indeed the Richardson number is used to identify unstable regimes also in strongly stratified flows and strong instabilities (and turbulence) can develop in the direction of the gravity, characterized a vertical Froude number**

of order _ 1 as described in Billant&Chomaz,"Self-similarity of strongly stratified inviscid flows," Phys. Fluids 13, 1645–1651 (2001).

We replaced the sentence line 2 page 4 by:

As well as the relationship between the integral scale with the Bulk Richardson number, this number provides a measure of the intensity mixing, and of how a turbulent flow is prone to develop instabilities. It is also used as a criterion for the existence or nonexistence of turbulence in a stably stratified environment (a large positive value of $Ri > 0.25$ is indicative of a decaying turbulence or a completely nonturbulent) (Arya, 2001).

**5 line 22 page 4. The text says: "..the integral scale of the mean wind direction u'". Here the authors probably means the horizontal direction but the way this is writtensounds wrong. Please rephrase it.**

We change the sentence line 22 23 page 4 by:

In this paper we focus on calculating the integral scales for horizontal and vertical component fluctuations u' and w'.

**7 line 1-2 page 6. See comment ## 2 and please specify better the way the datahave been processed. This is important for the readers to be able to reproduce (andtherefore validate) the results presented in the paper.**

We change the sentence line 1-2 page 6 by:

During the period of study the series are obtained in a consecutive period along the day every 30 minutes by a sonic anemometer and these series have been divided into five minutes series and over them the mean wind velocity of the horizontal and vertical components is calculated

**8 line 3-4 page 6. Authors claim they resolve vertical integral scales from 1m to1000m, which puzzles me a bit. Indeed, I am wondering how is it possible to detect(using the Taylor hypothesis) vertical scales larger than the hight(s) at which the sonicanemometer are placed? An answer to this point should be included in the report andand a compelling explanation integrated in the text.**

We have included the following explanation in line 5 of how vertical integral scales are possible to detect larger than the heights at which the anemometer are placed

We detect vertical scales over a broad range of scales from 1 m to 1000 m. The integral scales here are calculated based on the autocorrelation function, the mean wind velocity and integral time scale and each of them can be expected to vary significantly. As the integral scale are larger scales of turbulent flows is it possible to detect vertical scales larger than heights at which the sonic anemometer are located.

**9 line 20 page 6. Details on how the potential temperature has been estimatedshould be included in the text.**

We have included in the line 21 how the potential temperature has been estimated.

The potential temperature has been estimated as relative to ground level byusing the following formula: $\Delta\theta=\Delta T+\Gamma\Delta z$,  $\Gamma=0.0098$ K m$^{-1}$(Arya 2001)

**10 line 12 page 7. The text says: "Thus, stable stratification decreases the fractaldimension."Authors acknowledge they cannot provide rigorous arguments to explain the variationof the fractal dimension with the height. As a consequence the statement above soundsa bit too strong, unless they propose some solid argument to support it.**

As we cannot provide rigorous arguments to explain the variation of the fractal dimension  with the height. We removed from the text the statement"Thus, stable stratification decreases the fractal dimension."

**11 line 14-27 page 7. The text within these lines is poorly written and its meaning isa bit confusing. I suggest to re-written it from scratch and perhaps make it a bit moreconcise.**

We will  clarify the text within the line 14-27  page 6 by:

In Fig 4 it is observed how the integral scale varies versus time at the three heights. There are some questions that have not been clarified yet. For example: How doesthe diurnal and night cycle influence on the integral scale? Which is the mechanism responsible for the growth of the integral scale?It has been observed in the previous works that under certain conditions the turbulent flows self − organize and develop large-scale structures take place through an inverse cascade that occurs in stably stratified anisotropic flows (with or without rotation)(Marino et al., 2014, Smith and Waleffe, 2002). The inverse cascade mechanism might also be responsible for the growth of the integral scale in the stratified atmosphere. It is a fundamental issue that we should clarify in a future research. As it is indicated in Fig 4 the integral scale for u` component varies between around 100 m on their minor scales, until above 1500 m for its major scales. The integral scales for   w` component are slightly lower than u` component. It is shown that these vertical scales can reach sizes between a few tens of meters until 1000 m in some occasions. It is observed in each of the components that the greater is the height at which is located the anemometer, greater is the integral scale in turbulent flow. Usually, at 32 m these scales are, on average, greater than those of the 13 m and the latter higher than at 5.8 m height.

**12 There is a long standing debate in literature on whether an inverse cascade occursin stably stratified anisotropic flows (with or without rotation). The inverse cascademechanism, if any, might also be responsible for the growth of the integral scale in thestratified atmosphere. I thus suggest the authors to address this point,**

perhaps whenthey re-arrange the text (line 14-27) as indicated in the previous comment (## 11). Onthis note/topic I suggest to cite these two papers:
L. Smith and F. Waleffe, "Generation of slow large scales in forced rotating stratifiedturbulence," J. Fluid Mech. 451, 145–168 (2002).
R. Marino, P.D. Mininni, D. Rosenberg, and A. Pouquet, "Large-scale anisotropy instably stratified rotating flows". Phys. Rev. E 90, 023018 (2014).

This comment is appreciatedand the suggestions and the references of the two papers have been incorporated in the point (## 11)

**13 Conclusions should definitely be rearranged from line 16-24 to achieve a betterclarity of the text.**

We  clarify the text within the line 16-24 page 10 by:

We have calculated the fractal dimension and the integral scale of the horizontal and vertical components using wind velocity data at three different heights: 5.8 m, 13 m and 32 m. The numerical results show light significant differences on the diurnal and night cycle when the variation of the integral scale is analyzed versusthe fractal dimension. Atmospheric stratification is analyzed in the three heights through the Bulk Richardson number, finding thethree types of stratification during diurnal hours and at night hours the stable stratification. It would be interesting for future works to study the growth of the integral scale in stratified flows and if it could be due  to the inverse cascade on both diurnal and nighttime cycles. The main conclusions of this study are as follows.

**14 Units are missed in some plots, for instance in figure 2 "tau" is indicated but it isnot clear whether the lag is given in seconds, minutes or hours. Please double-checkall the figures and add the units (on the axes/legend) when it feels needed.**

Thank you for reminding the units in some plots.

Correct units in the following figures: Fig 2, tau(s), Fig 5, 6, 7, 8, 9 y 10 Integral scale (m).

---

## Author Comment (AC4) · 4 Aug 2016

**Response to the Referee #2**

We thank the reviewer 2 for the comments and the possible suggestions that can help to improve the paper. Although some of them we do not agree, below we respond point-by-point to the reviewers' comments (colored in blue)

Let me first recall that over the last 10 years, the statistical characterization of velocity turbulent fluctuations in the atmosphere is a widely studied topic. Numerous results from experiments and from numerical simulations are published in the literature.

It is true that there are numerous papers that have been published in the last 10 years over statistical characterization of velocity turbulent fluctuations in the atmosphere but not many of them establish the relation between the fractal dimension and the integral scale at changing the atmospheric stratification.

The fractal dimension, or box-counting dimension $D_0$, is a statistical property of turbulent signals, providing a measure of the degree of roughness, and it is thought to be a 'universal' property of it (see also many text books on the topic of the relation among the generalized fractal dimensions and the singularity spectrum of a turbulent signal).
This means, in particular, that it should not depend on the value of the integral scale of the flow, something that is clearly not universal but depends on the geometry of the system, and on its stability. If the authors differently think the such a relation exist, they should give a solid basis to their working hypothesis. A linear regression can be applied to any pair of observables.

We do not consider as a universal property the fractal dimension of u' and w' components of wind velocity, in fact we find that this magnitude depend on integral scale and stratification. This is deduced from the experimental data. During diurnal cycle the average values of the integral scale increase versus the fractal dimension for the u' horizontal component due to the instability of stratification into the ABL, and these values can be adjusted to the straight regression line with a $R^2$ acceptable. At nighttime hours due to the atmosphere stability the average values of the integral scale decrease with the increase in the fractal dimension. These values also fit a straight line as it is indicated in Fig 5 and Fig 6. For the w' vertical component have fitted a quadratic function as it is indicated in Fig 7. In all these cases the $R^2$ values and confidence levels are high. To find a theoretical background why this happens in the three heights is not easy.

The authors think that if the fractal dimension is a statistical property of turbulent signals providing a measure of the degree of roughness, the atmospheric stability of stratification somehow softens the irregular flow and instability increases such irregularity as is indicated in Fig 3. At night hours the fractal dimension decreases with an increase in noon as is observed in Fig 3.

The computation the fractal dimension is in principle an easy task, provided there existsan extended range of scales where there is scaling. The authors extract the fractaldimensions at fixed heights in the ABL with no discussion about the evolution of themixed layer, and kind of flow they expect to measure at different height depending

onthe stratification. For example: similar to homogeneous and isotropic 3D turbulenceduring convection, similar to homogeneous stratified flow at night, etc.. Even if thediurnal cycle is never really stationary, there are stages of quasi-stationarity when statisticalproperties can be assessed.

It is true that we do not discuss about the evolution of the mixed layer at fixed heights but that is not the objective of our work. The kind of flow measuring expected was that during the day there was instability due to sun radiation reaches the ground increasing the temperature of the lowest layer and during the nighttime hours expected stratified flows which it is confirmed by experimental data.

Finally, I find that the manuscript contains limited novelty with respect to Ref. "Tijera M.,Maqueda G.,Yaque C., and Cano J.: Analysis of fractal dimension of the wind speed and its relations with turbulent and stability parameters , Intech, Fractal Analisis 6 and Chaos in Geosciences, 29-46, 2012"

Although this work deal on the fractal dimension of the velocity components, there are significant differences with the cited paper because, both papers study different parameters of the turbulent flows. In the paper that you just quoted, the fractal dimension is analyzed versus the potential temperature, the turbulent kinetic energy, the friction velocity and Bulk Richardson number; but do not study the behavior of the wind fluctuations integral scale versus the fractal dimension at changing the atmospheric stratification nor separates hours of day and night to better analyze the influence of diurnal and nighttime cycle.

---

## Author Response (AR1)

**Response to the Referee #1**
**(Referee comments in black; response from the authors in blue)**

This paper investigates the possibility of correlations between the integral scale of stratified turbulent flows in the boundary layer and parameters characterizing topological features of the wind velocity field, such as the fractal dimension, and its stability properties studied through the Richardson number. Using data of the SABLES-98 campaign, at moderate temporal resolution, authors present here a series of results which provide interesting insights on the dynamics of the atmospheric flow in the site understudy. The outcome of the paper is relevant for the audience of Nonlin.Processes in along the lines detailed in the following.

We thank the referee for his positive reviews and constructive comments for our paper. We greatly appreciate the fact that our work is deemed acceptable for publication in NPG with some revisions. In the following, we respond point by point to the referee 1 remarks ( in blue).

**1 line 5-6 page 2. The text says: "The aim of this paper is to bridge the considerable gap that exists between the fractal dimension and the integral scale"**

We agree with referee #1 to clarify this sentence, and have it replaced by the following in this paper:

The aim of this paper is to investigate the possible correlations between the integral scale of the turbulent stratified flows in the Atmospheric Boundary Layer and parameters charactering topological features of the wind velocity field, such as fractal dimension and its stability properties, studied through the Bulk Richardson number. We are aware that there is a lack of investigations between the integral scale and fractal dimension.

**2 line 22-23 page 2. The text is not clear, is it a 5 minutes running average used to remove the mean field? This point needs to be better explicated in the text.**

We decided to change the sentence line 23-24 page 2 " In this paper we have carried out the necessary transformation to get the mean wind series in short intervals, namely 5 minutes ". It is not a 5 minutes running average. The new sentence reads as:

In this work, the series of wind velocities in the three directions x, y and z recorded by the anemometer are divided in series of non-overlapping five minutes length. Each of these series applies the necessary rotations to get the x-axis in the mean wind direction  (mean v is zero ) and zero mean vertical velocity (w vertical component)(Kaimal and Finningan, 1994).

**3 line 11-14 page 3. The sentence within these lines is not at all clear, please rephrase it.**

We rephrased the sentence line 11-14 page 3  by:

Turbulent motion of the atmospheric flows occurs through a broad range of scales from the smallest ones   that are usually defined as the scales at which the motion dissipates into heat due to the viscosity of the fluid until the larger scales   corresponding to the integral scale.

**4 line 2 page 4. The Richardson number provides a measure of how a turbulent flow is prone to develop instabilities. I thus disagree with referring to this parameter as a number that characterize the degree of stratification in the atmosphere. This definition (used in many places in the paper) is misleading and it is very important to modify it in the text not to drive the wrong message that stratified flows cannot be unstable. Indeed the Richardson number is used to identify unstable regimes also in strongly stratified flows and strong instabilities (and turbulence) can develop in the direction of the gravity, characterized a vertical Froude number**

of order _ 1 as described in Billant&Chomaz,"Self-similarity of strongly stratified inviscid flows," Phys. Fluids 13, 1645–1651 (2001).

We replaced the sentence line 2 page 4 by:

As well as the relationship between the integral scale with the Bulk Richardson number, which provides a measure of the degree of stability in the flow, and how this turbulent flow is prone to develop instabilities. It is also used as a criterion for the existence or non-existence of turbulence in a stably stratified environment (a large positive value over a critical threshold, is indicative of a decaying turbulence or a completely non-turbulent) (Arya, 2001).

**5 line 22 page 4. The text says: "..the integral scale of the mean wind direction u'". Here the authors probably means the horizontal direction but the way this is written sounds wrong. Please rephrase it.**

We change the sentence line 22 23 page 4 by:

In this paper we focus on calculating the integral scales for horizontal and vertical component fluctuations u′ and w'.

**7 line 1-2 page 6. See comment ## 2 and please specify better the way the data have been processed. This is important for the readers to be able to reproduce (and therefore validate) the results presented in the paper.**

We change the sentence line 1-2 page 6 by:

In this work, data from sonic anemometers measured at a sampling rate of 20 Hz installed at 5.8 (~ 6), 13 and 32 m are analyzed. 5-minute non-overlapping series are used to evaluate the different parameters.

**8 line 3-4 page 6. Authors claim they resolve vertical integral scales from 1m to 1000m, which puzzles me a bit. Indeed, I am wondering how is it possible to detect(using the Taylor hypothesis) vertical scales larger than the hight(s) at which the sonic anemometer are placed? An answer to this point should be included in the report and a compelling explanation integrated in the text.**

We have included the following explanation in line 5 of how vertical integral scales are possible to detect larger than the heights at which the anemometer are placed:

We detect vertical scales over a broad range of scales from 1 m to 1000 m. The integral scales here are calculated based on the autocorrelation function, the mean wind velocity and integral time scale, and each of them can be expected to vary significantly. As the integral scale are the larger scales of turbulent flows it is possible to detect vertical scales larger than heights at which the sonic anemometer are located.

**9 line 20 page 6. Details on how the potential temperature has been estimated should be included in the text.**

We have included in the line 21 how the potential temperature has been estimated.

The potential temperature has been estimated as relative to ground level by using the following formula: $\Delta\theta=\Delta T+\Gamma\Delta z$, $\Gamma=0.0098$ K m$^{-1}$(Arya 2001)

**10 line 12 page 7. The text says: "Thus, stable stratification decreases the fractal dimension."Authors acknowledge they cannot provide rigorous arguments to explain the variation of the fractal dimension with the height. As a consequence the statement above sounds a bit too strong, unless they propose some solid argument to support it.**

As we cannot provide rigorous arguments to explain the variation of the fractal dimension with the height, we have removed from the text the statement"Thus, stable stratification decreases the fractal dimension."

**11 line 14-27 page 7. The text within these lines is poorly written and its meaning is a bit confusing. I suggest to re-written it from scratch and perhaps make it a bit more concise.**

We will clarify the text within the line 14-27 page 7 by:

In Fig 4 it is observed how the integral scale varies versus time at the three heights. There are some questions that have not been clarified yet in the literature. For example: How does the diurnal and night cycle influence on the integral scale? Which is the mechanism responsible for the growth of this integral scale? It has been observed in previous works that under certain conditions the turbulent flows self − organize and develop large-scale structures that take place through an inverse cascade that occurs in stably stratified anisotropic flows (with or without rotation) (Smith and Waleffe, 2002, Marino et al., 2014). The inverse cascade mechanism might also be responsible for the growth of the integral scale in the stratified atmosphere. It is a fundamental issue that should be clarified in a future research. As it is indicated in Fig 4 the integral scale for u` component varies between around 100 m on their smaller scales, until above 1500 m for its larger scales. The integral scales for w` component are slightly lower than for u` component. It is shown that these vertical scales can reach sizes between a few tens of meters until 1000 m in some occasions. It is observed, for each of them, that the greater is the height at which is located the sonic, the greater is the integral scale in the turbulent flow. Usually, at 32 m these scales are, on average, greater than those of the 13 m and the latter higher than at 5.8 m height.

**12 There is a long standing debate in literature on whether an inverse cascade occurs in stably stratified anisotropic flows (with or without rotation). The inverse cascade mechanism, if any, might also be responsible for the growth of the integral scale in the stratified atmosphere. I thus suggest the authors to address this point, perhaps when they re-arrange the text (line 14-27) as indicated in the previous comment (## 11). On this note/topic I suggest to cite these two papers:**
L. Smith and F. Waleffe, "Generation of slow large scales in forced rotating stratified turbulence," J. Fluid Mech. 451, 145–168 (2002).

R. Marino, P.D. Mininni, D. Rosenberg, and A. Pouquet, "Large-scale anisotropy instably stratified rotating flows". Phys. Rev. E 90, 023018 (2014).

This comment is appreciated and the suggestions and the references of the two papers have been incorporated in the point (## 11)

**13 Conclusions should definitely be rearranged from line 16-24 to achieve a betterclarity of the text.**

We clarify the text within the line 16-24 page 10 by:

We have calculated the fractal dimension and the integral scale of the horizontal and vertical components using wind velocity data from sonic anemometers at three different heights: 5.8 m, 13 m and 32 m. The numerical results show light significant differences on the diurnal and night cycle when the variation of the integral scale is analyzed versus the fractal dimension. Atmospheric stratification is analyzed for the three heights through the Bulk Richardson number, finding the three classical types of stratification along the diurnal cycle. It would be interesting for future works to study the growth of the integral scale in stratified flows and if it could be due to the inverse cascade on both diurnal and nighttime cycles. The main conclusions of this study are as follows.

**14 Units are missed in some plots, for instance in figure 2 "tau" is indicated but it is not clear whether the lag is given in seconds, minutes or hours. Please double-checkall the figures and add the units (on the axes/legend) when it feels needed.**

Thank you for reminding the units in some plots. We have solved these missed units for the different figures.

**Response to the Referee #2**

We thank the reviewer 2 for the comments and the possible suggestions that can help to improve the paper. Although some of them we do not agree, below we respond point-by-point to the reviewers' comments (colored in blue)

Let me first recall that over the last 10 years, the statistical characterization of velocity turbulent fluctuations in the atmosphere is a widely studied topic. Numerous results from experiments and from numerical simulations are published in the literature.

It is true that there are numerous papers that have been published in the last 10 years over statistical characterization of velocity turbulent fluctuations in the atmosphere but not many of them establish the relation between the fractal dimension and the integral scale at changing the atmospheric stratification, which is the objective of the present paper.

The fractal dimension, or box-counting dimension $D_0$, is a statistical property of turbulent signals, providing a measure of the degree of roughness, and it is thought to be a 'universal' property of it (see also many text books on the topic of the relation among the generalized fractal dimensions and the singularity spectrum of a turbulent signal).
This means, in particular, that it should not depend on the value of the integral scale of the flow, something that is clearly not universal but depends on the geometry of the system, and on its stability. If the authors differently think the such a relation exist, they should give a solid basis to their working hypothesis. A linear regression can be applied to any pair of observables.

We do not consider as a universal property the fractal dimension of u' and w' components of wind velocity, in fact we find that this magnitude depend on integral scale and stratification. This is deduced from the experimental data. During diurnal cycle the average values of the integral scale increase versus the fractal dimension for the u' horizontal component due to the instability of stratification into the ABL, and these values can be adjusted to the straight regression line with a $R^2$ acceptable. At nighttime hours due to the atmosphere stability the average values of the integral scale decrease with the increase in the fractal dimension. These values also fit a straight line as it is indicated in Fig 5 and Fig 6. For the w' vertical component have fitted a quadratic function as it is indicated in Fig 7. In all these cases the $R^2$ values and confidence levels are high. Find a theoretical background of why this happens at the three heights is not easy, but the fact to show it can be interesting for the NPG readers.

We also think that if the fractal dimension is a statistical property of turbulent signals providing a measure of the degree of roughness, the atmospheric stability somehow softens the irregular flow and instability increases such irregularity as is indicated in Fig 3. At night hours the fractal dimension decreases while it increase in noon as is observed in Fig 3.

The computation the fractal dimension is in principle an easy task, provided there exists an extended range of scales where there is scaling. The authors extract the fractal dimensions at fixed heights in the ABL with no discussion about the evolution of the mixed layer, and kind of flow they expect to measure at different height depending on the stratification. For example: similar to homogeneous and isotropic 3D turbulence during convection, similar to homogeneous stratified flow at night, etc.. Even if the diurnal cycle is never really stationary, there are stages of quasi-stationarity when statistical properties can be assessed.

It is true that we do not discuss about the evolution of the mixed layer but this is out of the scope of our work. With regards to the kind of flow we expect to measure at different height, we have to say that during diurnal hours it is expected to have a flow of similar characteristics, while at night, especially for strong stable stratification, it can exist different degree of stability at these 3 heights affecting then to the flow characteristics. However these topics are also out of the scope of the present work, where we want to underline the relationships between integral sales and fractal dimension and also between integral scales and the degree of stability, evaluated from the bulk Richardson number.

Finally, I find that the manuscript contains limited novelty with respect to Ref. "Tijera M.,Maqueda G.,Yaque C., and Cano J.: Analysis of fractal dimension of the wind speed and its relations with turbulent and stability parameters , Intech, Fractal Analisis 6 and Chaos in Geosciences, 29-46, 2012"

Although this work deal on the fractal dimension of the velocity components, there are significant differences with the cited paper because, both papers study different parameters of the turbulent flows. In the paper that you just quoted, the fractal dimension is analyzed versus the potential temperature differences, the turbulent kinetic energy, the friction velocity and Bulk Richardson number; but do not study the behavior of the wind fluctuations integral scale versus the fractal dimension at changing  atmospheric stratification neither separates its diurnal cycle (including the nocturnal  behaviour) to better analyze and understand the influence of diurnal and nocturnal hours.

[revised manuscript text omitted]

$1.2 \ 10^3$ , (b) $L_{intu}(13 \text{ m})= -1 \ 10^3 \ DF_u(13 \text{ m}) + 1.37 \ 10^3$, (d) $L_{intu}(32 \text{ m})= -1.4 \ 10^3 \ DF_u(32$
m) $+ 1.9 \ 10^3$

[Figure]

**Figure 7.** Variations of the integral scale versus the fractal dimension of the w` component at 5.8 m, 13 m and 32 m. (a), (c) y (e) diurnal hours, (b), (d) y (f) night. hours. The fits to a quadratic function of the averages values appear on the top left of the each graph, being y variable $L_{intw}$ and x variable $DF_w$.

[Figure]

**Figure 8.** Integral length scales of u` component plotted against the Bulk Richardson number at 5.8 m,  13 m and 32 m. (a), (c) y (e) diurnal hours, (b), (d) y (f) night hours.

[Figure]

**Figure 9.** Integral length scales of w` component plotted against the Bulk Richardson
number at 5.8 m,  13 m and 32 m. (a), (c) y (e) diurnal hours, (b), (d) y (f) night hours

[Figure]

**(a)**

**(b)**

**(c)**

**Figure 10.** Comparison of integral scales of u` component and w` component (a) at 5.8 m, (c) at 13 m (c) at 32 m, showing that the averages values of theses scales fit to the linear regression indicated on the top left each graph. Data set appear as a cluster around the straight line.